# Distance to Natural Environments, Physical Activity, Sleep, and Body Composition in Women: An Exploratory Analysis

**DOI:** 10.3390/ijerph20043647

**Published:** 2023-02-18

**Authors:** Andreia Teixeira, Ronaldo Gabriel, José Martinho, Irene Oliveira, Mário Santos, Graça Pinto, Helena Moreira

**Affiliations:** 1Department of Sports Science, Exercise and Health, University of Trás-os-Montes and Alto Douro, 5000-801 Vila Real, Portugal; 2Centre for the Research and Technology of Agro-Environmental and Biological Sciences (CITAB), Institute for Innovation, Capacity Building and Sustainability of Agri-Food Production (Inov4Agro), University of Trás-os-Montes and Alto Douro, 5000-801 Vila Real, Portugal; 3Department of Geology, Geociencies Centre (CGeo), University of Trás-os-Montes and Alto Douro, 5000-801 Vila Real, Portugal; 4Department of Mathematics, University of Trás-os-Montes and Alto Douro, 5000-801 Vila Real, Portugal; 5Center for Computational and Stochastic Mathematics (CEMAT-IST), Instituto Superior Técnico, University of Lisbon, 1049-001 Lisbon, Portugal; 6Laboratory of Fluvial and Terrestrial Ecology, Innovation and Development Center, University of Trás-os-Montes e Alto Douro, 5000-801 Vila Real, Portugal; 7Laboratory of Ecology and Conservation, Federal Institute of Education, Science and Technology of Maranhão, Rua do Comercio, 100, Buriticupu 65393-000, MA, Brazil; 8Center in Sports Sciences, Health Sciences and Human Development (CIDESD), University of Trás-os-Montes and Alto Douro, 5000-801 Vila Real, Portugal

**Keywords:** women, nature exposure, actigraphy parameters, adiposity levels

## Abstract

A growing body of evidence indicates that living close to nature is associated with better health and well-being. However, the literature still lacks studies analyzing the benefits of this proximity for sleep and obesity, particularly in women. The purpose of this study was to explore how distance to natural spaces is reflected in women’s physical activity, sleep, and adiposity levels. The sample consisted of 111 adult women (37.78 ± 14.70). Accessibility to green and blue spaces was assessed using a geographic-information-system-based method. Physical activity and sleep parameters were measured using ActiGraph accelerometers (wGT3X-BT), and body composition was assessed using octopolar bioimpedance (InBody 720). Nonlinear canonical correlation analysis was used to analyze the data. Our findings reveal that women living in green spaces close to their homes had lower levels of obesity and intra-abdominal adiposity. We also demonstrated that a shorter distance to green spaces seemed to correlate with better sleep onset latency. However, no relationship was found between physical activity and sleep duration. In relation to blue spaces, the distance to these environments was not related to any health indicator analyzed in this study.

## 1. Introduction

Women play crucial roles within families and society as caregivers, educators, wives, mothers, and members of the community [1]. Despite their important contributions, women often face worse health outcomes than men, which highlights their status as a marginalized and vulnerable group in society. According to Eurostat [2] and the data published by the European Institute for Gender Equality (EIGE), Portuguese women have a higher average life expectancy than men (84.1 years vs. 78 years, respectively), but they also have a lower estimated life quality (2.1 years less) compared to men. The fact that women in Portugal have a longer life expectancy than men does not necessarily mean that their overall health is better. The reason for the difference in life expectancy could be due to a variety of factors, including biological differences, access to healthcare, and lifestyle factors. For example, women may be more likely to experience certain conditions, such as osteoporosis, and may face unique health challenges related to pregnancy, childbirth, and menopause. Women also experience higher rates of depression, anxiety, and other mental health conditions [3,4,5].

Women’s health and well-being also correlate with factors such as physical activity, sleep, and adiposity levels, which are of critical importance to study. According to the latest Eurobarometer on Sport and Physical Activity [6], only 20% of women in Portugal engage in physical exercise or sports activities, and these rates decline with age, particularly among women over 55. As reported by Rossi, et al. [7], postmenopausal women tend to have lower levels of moderate- and vigorous-intensity physical activity (PA) and higher levels of light-intensity PA.

In relation to sleep, women typically report poorer quality and more disrupted sleep across various stages of life [8]. Regarding sleep duration, men tend to sleep less than women on average in all age groups, and this difference is based on both biological and social factors [9]. In fact, physiological and hormonal changes that occur during puberty, the menstrual cycle, pregnancy, and menopause can impact women’s sleep architecture and quality [8,10]. Sleep is vital for health and well-being in all age groups and both sexes [11]. It is important for cognitive function, mood, mental health, and cardiovascular, cerebrovascular, and metabolic health [12]. However, there is currently limited knowledge on the association between sleep patterns and living near natural spaces [13,14] and for that reason, we objectively assessed this variable in the present study.

Regarding body composition, in the literature, the differences between genders are well established [15,16]. Women tend to have a higher percentage of fat mass and a lower muscular condition compared to men. Despite the fact that total and central adiposity peaks are reached after the age of 50 years and 60 years in women and men, respectively [17], this increase is especially evident in women due to menopause [18]. Depletion of estrogen during menopause generates an increase in fat mass, resulting in the extravasation of excessive lipid quantities and production of inflammatory cytokines [19], promoting the ectopic deposition of fat in the muscle and various organs.

Regarding the benefits of natural environments, numerous studies have emphasized their significance in improving health outcomes [20,21]. Green spaces, defined as areas dominated by vegetation such as grass, trees, shrubs, and more, including urban and suburban forests, parks, community gardens, and even school yards, have been found to have a positive impact on health [22]. Blue spaces, also referred to as blue infrastructure, consist of all areas dominated by surface water bodies or watercourses [22].

The mechanisms behind the benefits of natural environments include increasing physical activity, reducing psychological stress, promoting social cohesion and interaction, and reducing exposure to urban environmental hazards such as air pollution, noise, and heat islands [23,24,25]. These positive effects are known to contribute to a healthy weight [24,26], as well as to improving sleep duration and quality [27]. It can be speculated that individuals with access to more natural spaces are more likely to exhibit better health outcomes as a result of these positive drivers.

Another important aspect of natural spaces is the presence of biodiversity. The level of biodiversity can serve as an indicator of environmental quality and has been linked to improved health outcomes [28], increased psychological well-being [29], and positive emotions [30].

Many studies from which current research evidence is drawn have several important limitations. In general, objective measures of physical activity and sleep are not included and these variables are typically assessed through questionnaires [27,31,32,33]. In the vast majority of past studies, the assessment of total adiposity levels was performed mainly through self-reported weight and height values [34]. Additionally, few studies have differentiated between green and blue spaces. Völker and Kistemann [35] state that the term "blue space" summarizes all visible surface waters in the environment as an analogy to green spaces, rather than as a sub-category. They argue that while green spaces provide diverse forms of perception, such as changes in seasons and the variety of flora and fauna, they cannot attain the same symbolic semantic influence as water. The authors also note that contemplation in green spaces is not as pronounced as in blue spaces, making it important to consider these environments separately.

Several reasons support the relevance of developing studies in the field of natural spaces that involve women exclusively. Firstly, differences in physiology, psychology, nurturing styles, and thinking styles between men and women may impact the aspects of the residential environment that they focus on [36]. Secondly, women experience worse health outcomes than men, often report chronic pain [37], and face more barriers to weight loss [38]. The results of Hologic Global Women’s Health Index [39] show that women in Portugal have a lower overall health and well-being score of 58 points. This index is a comprehensive measure that takes into account key indicators such as access to preventive care, emotional well-being, perceptions of health and safety, and basic needs and personal health. In Portugal, the scores of the five individual indicators range from 37 points for preventive care to 84 points for perceptions of health and safety. The index is scored on a scale of 0 to 100, with a higher score indicating a better overall experience for women in these areas. Additionally, women continue to be responsible for most household chores, childcare, and caregiving, and thus spend more time in the residential environment [40,41]. Furthermore, as reported by Loarne-Lemaire, et al. [42], women are also more vulnerable to climate problems due to social, economic, and cultural factors, making them more sensitive to environmental challenges. Lastly, despite not always having easy access to green spaces, living in close proximity to green spaces, having green spaces with features or amenities that are particularly appealing to their needs and preferences, or feeling safe while using them, women still tend to benefit from natural environments more than men [43,44].

Thus, the aim of this study was to investigate the association between the distance to natural spaces, physical activity, sleep, and adiposity levels in adult women.

## 2. Materials and Methods

### 2.1. Study Area

This study was carried out in north Portugal, within the county of Vila Real (Figure 1). The study area presents a Csb type of climate, i.e., a temperate climate with a dry or temperate summer [45]. The landscape is characterized by continuous and discontinuous urban areas intercalated with agricultural patches, forests, and semi-natural vegetation.

Blue spaces, mostly lotic environments, belong to the Corgo River Basin, a tributary on the right bank of the Douro River, which has the largest river catchment in the Iberian Peninsula. The riparian forests of blue spaces, associated with alder (*Alnus glutinosa*), ash (*Fraxinus angustifolia*), willow (*Salix* sp.), and thickets and nettle (*Celtis australis*), encompass several uncommon and confined plant species [46]. Additionally, these habitats are particularly biodiverse and important for the conservation of endangered endemic aquatic animals. Concerning green spaces, remnant oak forests (*Quercus* sp.) and heathlands (*Erica* sp. and *Calluna* sp.) are particularly relevant for their floristic biodiversity and the occurrence of species of conservation concern (e.g., *Veronica micrantha*). Anyhow, in the location of the study area, which is in the transition from the “Atlantic” to the “Mediterranean” biogeographic regions, even highly humanized areas such as planted forests, vineyards, pastures, cropland, and vegetable and public gardens contain a diversity of intermingled species and habitats [47].

### 2.2. Ethics Statement

This research was approved by the Ethics Committee of the University of Trás-os-Montes and Alto Douro (Reference No.: Doc51A-CE-UTAD-2020) and adhered to the guidelines set forth in the Declaration of Helsinki. Measures were taken to prevent the transmission of COVID-19 during the study. Participants were fully informed of the purpose, benefits, and potential risks of the study and provided written informed consent.

### 2.3. Study Design and Sample

This cross-sectional study was conducted between December 2020 and February 2021 and included a sample of 111 adult women with an average age of 37.78 ± 14.70 years. The participants were not using any medication that could impact their sleep.

### 2.4. Exposure Assessment

Distance to natural spaces: “Green space” included open, accessible, available recreational and sustainable spaces, in the form of parks, wetlands, conservation reserves, and sports fields, and could comprise forestland, pastureland, or other natural areas at least 1 ha in size. Both public and private gardens were considered green space areas [48,49,50]. Agricultural land was not included because it is not freely accessible. Green space was considered accessible when there is a road, footpaths, trails, walkways, within or near the area (no more than 25 m). Regarding “blue space”, this was defined as an outdoor environment—either natural or manmade—that prominently featured water and was accessible to humans either proximally (being in, on, or near water) or distally/virtually (being able to see, hear, or otherwise sense water) [51]. Addresses were geocoded using the Geocode figure tool with the Esri World Geocoder (ArcGIS Pro). The road network used was the Open Street Map for Portugal, after updating it. The street network was constructed using the Create Network Dataset with the Road Centerlines tool (ESRI, 2022). The green areas were converted to a 5 m pixel raster. These images were later converted into points (at the center of the pixel) to allow the calculation of the distances between points along a street network. Distances were calculated using the Closest Facility tool (Version 3.0, ArcGIS Pro, ESRI, Redlands, CA, USA). In this study, distances of less than 300 m to green spaces and less than 500 m to blue spaces were considered indicative of good access to these environments, based on recommendations from the United Nations [52] and other studies [53,54,55,56].

### 2.5. Health Measures

Body composition: Body height (BH) was measured using a stadiometer (SECA 220, Seca Corporation, Hamburg, Germany). Body mass (BM, kg), fat mass (FM, kg and %), visceral fat area (VFA, cm^2^), and appendicular skeletal muscle mass (ASMM, kg) were evaluated using the octopolar bioimpedance InBody 720 (Biospace, Seoul, South Korea), with an alternating multifrequency of 1, 5, 50, 250, 500, and 1000 kHz.

This technology employs eight contact electrodes; two are positioned on the palm and thumb of each hand and the other is placed on the front part of the feet and on the heels. According to the criteria specified in the equipment manual [57,58], participants were instructed to (1) not eat food for at least 4 h; (2) not perform moderate-to-vigorous physical activity 12 h before the evaluation; (3) use the bathroom 30 min before the test (to reduce the volume of urine and feces); (4) not consume alcoholic beverages for at least 48 h; and (5) not wear metal jewelry. Before contact with the electrodes, the participants cleaned their hands and feet with antibacterial tissue obtained from the manufacturer. The data were electronically imported into spreadsheets using the software Lookin’Body 120 (Biospace, Seoul, South Korea).

The appendicular skeletal muscle mass index (ASMMI = ASMM/BH^2^, kg/m^2^) was used to categorize the muscle condition, and the cut-off value for low muscle mass was <5.7 kg/m^2^ [59]. The cut-off points for elevated visceral fat area and obesity were as follows: VFA ≥ 100 cm^2^ [60] and FM ≥ 35% in women [61].

Considering dual-energy X-ray absorptiometry as a reference method, several studies have documented the validity of this equipment in assessing FM (kg and %) and ASMM in adults and older people [62,63,64,65,66]. Several authors have also reported a significant association between VFA estimated using the equipment and that measured using computerized axial tomography [67,68].

Physical activity and sleep: A triaxial accelerometer (ActiGraph GT3X, Actigraph Inc., Pensacola, FL, USA) was used to assess bodily movements 24 h per day for four consecutive days (two weekdays and weekends), including sleep outcomes, and was worn on the non-dominant wrist. Each participant was instructed to remove the device only when engaging in water-based activities (e.g., showering/bathing or swimming). All participants who agreed to wear the accelerometer received standardized oral and written information about using the equipment on the study days. The subjects also filled out a record sheet where they reported their sleep time and non-wear time.

We used a sampling rate of 100 Hz and a 1 min epoch setting. A valid wear day consisted of a device wear time of at least 600 min. The start of the devices was programmed for 6 am on the first day of evaluation and the physical activity records considered 15 s periods. Non-wear time was defined as 90 consecutive minutes of zero counts, with an allowance of 2 min of nonzero counts, provided there were 30 min of consecutive zero-count windows up- and downstream [69].

Activity intensities were then classified using counts per minute (CPM) thresholds: moderate: 1952–5724 CPM and vigorous: ≥5725 CPM [70]. The recommended PA level was at least 150–300 min/week of moderate intensity activity [71]. The number of daily steps was also retrieved from the accelerometer. Sleep patterns were assessed using a previously validated software algorithm based on the Cole–Kripke scoring method [72]. For this study, we describe the following objective sleep measures: total sleep time (TST, hours), sleep onset latency (SOL, min), sleep efficiency (SE, %), fragmentation (SFI, %), sleep onset time, sleep offset time, and sleep midpoint time. The TST is the amount of sleep obtained at night as identified by ActiLife software (hours/night). Sleep duration was categorized into sufficient sleep (≥7 h/night) versus insufficient sleep (<7 h/night), based on the study of Watson et al. [12]. The SOL is the duration of the time between when the lights are turned off and the individual tries to sleep until the moment he/she actually falls asleep and was determined by the ActiLife software. The cut-off points considered for SOL was 30 min [73]. SE is the percentage of the sleep period spent sleeping (duration of sleep/duration of time in bed) in percentage format. SE was categorized into low (<85%) or high efficiency (≥85%) [74]. Sleep fragmentation was evaluated as an index that tabulates the frequency of mobility episodes and short sleep bouts between sleep onset time and sleep offset time. It is calculated as the sum of percent mobile and percent 1 min immobile bouts divided by the number of immobile bouts [75].The higher the SFI, the more sleep is disrupted [76]. The sleep onset time and sleep offset time correspond to the first and the last epoch scored, respectively, as sleep in clock time format (HH:MM). The sleep midpoint is clock time that represents the midpoint between the clock time of sleep onset and clock time of sleep offset and was calculated as sleep onset time + ((wake up time-sleep onset time)/2) [77]. Based on a cut-off point for healthy sleep timing per published data, the sleep midpoint variable was then dichotomized as healthy (occurring between 2:00 and 4:00 AM) or early/late (outside of 2:00–4:00 AM) [78,79]. Data were evaluated with ActiLife software (Version 6.13.4, Pensacola, CA, USA).

### 2.6. Demographic Characteristics

According to a prior literature review, we evaluated several demographic variables including age, marital status, and employment status. We also gathered information about the consumption of alcohol and tobacco, as well as the intake of coffee, tea, and caffeine-containing beverages, since these have an impact on sleep.

### 2.7. Statistical Analyses

The numeric data were presented as mean ± standard deviation and qualitative variables were presented as absolute frequencies and percentages. To assess the relative contributions of the accessibility of green and blue spaces, nonlinear canonical correlation analysis (OVERALLS) was used. The OVERALS technique is a nonlinear multivariate exploratory analysis used to deal with variables with different levels of measurement, such as numerical, ordinal, and nominal levels, and that are defined by at least one set of variables. The purpose is to determine how similar these sets of variables are and how the projection of the data in a low-dimensional space can be achieved, accounting, as much as possible, for the variance in the relationships among the sets, and at same time to establish the similarities between the sets. In OVERALS, as in principal component analysis (PCA), eigenvalues are associated with each dimension and indicate to what extent every single dimension accounts for a good fit of data in a low-dimension space and, in a centroid plot, score objects for categories of each variable that are nearest to each other, indicating a higher degree of similarity [80]. We used 13 variables and classified them into 3 sets: (1) distance to green and blue space; (2) accelerometry variables (MVPA and sleep); and (3) body composition variables (%FM and VFA). The labels of the sets, variables, and categories in data, and the symbols that represent the categories in the graphics, are given in Appendix A. All analyses were carried out using IBM SPSS, version 27.0 (Chicago, IL, USA).

## 3. Results

A data description is presented in Table 1 and Table 2. The mean age of the sample was 37.78 ± 14.70 years. A total of 0.52.3% of the participants were single and 61.3% were employed. Regarding the factors that influence sleep, 53.2% of the women drank between 1 and 2 cups of coffee a day, some drank alcohol sometimes (65.8%), and most of them did not have a smoking habit (87.4%).

Overall, the majority of respondents had good access to green space and blue space, with 40.5% and 67.6% of people living within 300 m of green and 500 m of blue space, respectively.

Regarding body composition, 39.6% of the women were classified as obese, with over 25% of the sample revealing high levels of intra-abdominal adiposity. Based on the ASMMI values, low muscle mass was identified only in 10 women. The mean MVPA (188.03 ± 118.99 min/week) and steps-per-day values (12712.98 ± 3816.90 steps) identified were within the values recommended in the literature, with about half of the women (55.9%) being classified as physically active. Additionally, 58.6% of the sample had a sleep duration within the recommended range of 7 to 8 h [73,81]. Concerning sleep quality, the sample exhibited efficient sleep, 48.6% took less than 30 min to fall asleep (latency), and 82.9% showed reduced values of the sleep fragmentation index. The average sleep midpoint was 4:19 AM ± 2 h, representative of late sleep chronotypes. 

As shown in Figure 2, about half of the women (51.1%) who lived within 300 m of green spaces did not reach the recommended levels of MVPA. For blue spaces, the prevalence of recommended and non-recommended levels of MVPA was similar for both distances analyzed. A large percentage (68.9%) of women living near green spaces exhibited normal total adiposity values. In contrast, a high proportion (69.4%) of obese women were registered as living less than 500 m from blue spaces. Concerning intra-abdominal adiposity levels, a higher percentage (75.6%) of women with normal VFA levels were observed to live close to green spaces compared to those living further away (68.2%). In opposition, there was a higher prevalence of women with high central adiposity levels living closer to blue spaces (83.3%). Of the women who lived closer to green spaces, 47% slept at least 7 h per night, while among those who lived further away, 66.7% exhibited non-recommended hours of sleep. For blue spaces, there was an equitable distribution of recommended and non-recommended TST. A higher prevalence of women living within 300 m of green spaces had recommended sleep onset latency levels compared to women who lived the farthest away (54.5% vs. 45.5%). Those living closer to blue spaces took longer to fall asleep compared to those with recommended levels of sleep onset latency (53.3% vs. 46.7%).

We consider that distance to natural spaces is not an isolated factor that can influence the duration and quality of sleep. Sleep is affected by various social, psychosocial, and environmental factors. Stress [82], socioeconomic status [83], cell phone habits [84], and distance to workplace [85], for example, are known contributors to poor sleep, but these were not the subject of the current study.

Loss values, eigenvalues, and fit values to show the similarities among sets are presented in Table 3. The eigenvalues obtained from this study were 0.427 (first dimension) and 0.403 (second dimension). The eigenvalues of the two dimensions add up to a fit of 0.830, and they can be interpreted as a proportion of explained variance. The fit shows to what extent the nonlinear canonical correlation analysis solution fits the optimally quantified data in regard to the association between the sets. We used two-dimensional solutions, so 0.830/2 = 41.5% of the variation was calculated in the analysis.

The component loadings presented in Figure 3 give the correlations between object scores and optimally scaled variables, and display the coordinates of the variable points on the graph. Component loadings indicated that visceral fat area and sleep onset latency were the most effective variables in relationships among variable sets, because they were positioned far away from the origin. The other variables (FM, MVPA, and TST) had no intense effect on relationships, because they were positioned close to the origin.

The plot of centroids, labeled by categories, is presented in Figure 4. The results show that living within 300 m of green spaces was associated with recommended levels of sleep onset latency (SOL_R) and lower levels of fat mass (FM_L). Women who lived closer to green spaces tended to have a shorter time transitioning from wakefulness to sleep and lower levels of total body fat. On the other hand, the results show that living near blue spaces (within 500 m) was associated with non-recommended levels of physical activity (MVPA_NR) and sleep duration (TST_NR). This suggests that proximity to blue spaces does not necessarily lead to adequate levels of physical activity and sleep duration.

## 4. Discussion

The purpose of this study was to explore the association between the distance to natural spaces and levels of physical activity, sleep duration and quality, and adiposity levels. According to the last Eurobarometer on Sport and Physical Activity [6], for most European Union countries (47%), including Portugal (52%), natural environments are valued for practicing physical activity (PA), and one of the main reasons reported by Europeans for practicing PA is to improve health. These results support the importance of studying the distance to these natural environments as an indication of health.

Our findings reveal that women with green spaces near their homes showed lower levels of total and intra-abdominal adiposity and better sleep onset latency values. However, we found no relationship between green space with physical activity and sleep time. Regarding blue spaces, the distance to these environments did not relate to any health indicator analyzed in this study (Figure 5).

The methods used to assess exposure to nature vary widely, and typically can be broadly classified into three categories: availability, distance, and visibility [86]. The measurement of proximity to natural environments is often assessed through geographical information systems (GIS), which correspond to the distance by road from the residential location to the nearest natural area [87]. This technology makes it possible to assess the quantity and quality of environments through remote sensing images. There are already recommendations for the distance between homes and the nearest open public space. Nonetheless, these are still not widely accepted. According to the European Commission [88], the currently recommended distance between a residence and the nearest open public space is 300 m. This recommendation might be supported by the fact that 300–400 m is the threshold after which the use of green spaces starts to decline [89]. Konijnendijk [90] recommends that every citizen be able to see three trees, live in a 30% green-covered neighborhood, and be within 300 m of a park. As reported by the same author, benefits thus appear to result from both a combination of the provision in terms of proximity and size or coverage of natural environments. A better understanding of the minimum distance to consider in analyzing green space would help urban planners and public health professionals to improve the way they design green spaces close to the community [91].

In our study, we found that distance to green or blue spaces was not related to levels of moderate-to-vigorous physical activity (MVPA), which contradicts results from previous studies [56,92,93,94,95,96]. It is important to note that in our research, both MVPA and distance to natural spaces were objectively measured, while in prior studies, the distance was typically self-reported. According to Schipperijn, et al. [97], self-reported distances tend to be better predictors of the frequency of use of natural spaces than objectively measured distances, which may bias results. In addition, in the studies developed for several authors [56,93,94,95]. the levels of physical activity were self-reported, which is often considered as an important limitation. Additionally, many of the previous studies analyzed both genders without differentiating between the physical activity levels of men and women.

Our contrasting findings could also be related to the fact that previous studies used various definitions of green and blue spaces (as seen in Appendix A) and employed different methods to evaluate accessibility to these environments, such as Euclidean or network buffers, average distances, or the distance to the nearest natural environment from a residence. These differences make it difficult to compare results across studies [97,98,99]. Additionally, in our study, we considered a green space to be accessible if there was a road, footpath, trail, or walkway within or near (no further than 25 m) the area. However, this type of public accessibility is not always ensured in other studies.

On the other hand, our results may suggest that living close to natural spaces is not enough to influence levels of physical activity; perhaps it is the enjoyment of these spaces that increases these levels. Russell, et al. [100] propose that benefits derived from interactions with natural environments may be obtained through four different channels of human experience: *knowing* (metaphysical interactions that arise through thinking about an ecosystem, its components, or the concept of an ideal ecosystem, in the absence of immediate sensory inputs); *perceiving* (remote interactions with ecosystem components, often associated with visual information alone; *interacting* (physical, active, direct multisensory interactions with ecosystem components that may be cursory and may involve other people); and *living within* (the everyday, repetitive, pervasive, voluntary, or involuntary interaction with the ecosystem in which one lives).

The lack of associations may be due to the fact that women lack opportunities or feel unsafe accessing green and blue space. Several investigations document that women attribute a higher value to quality and security features of green spaces than men, leading to increased use by women in safer and higher-quality natural environments [101]. In our study the predominant green spaces were forestland, pastureland, and similar green areas, which may possibly not be safe for women to use. Braçe et al. [40] highlight that the main differences in men’s and women’s perception of natural environments include lighting, safety, cleanliness, walking routes, bike lanes, shaded areas, recreational areas, off-leash dog areas, playgrounds, drinking fountains, and pleasant views.

When we investigated the relationship between distance to natural spaces and body composition, particularly total and central adiposity levels, we identified numerous gaps in the literature. Firstly, there are limited studies on blue spaces [102,103]. Secondly, many studies do not differentiate between green and blue spaces, often including both in their definition of green spaces. Thirdly, most studies on this topic define obesity by using total adiposity levels. We only found one study that evaluated central adiposity using waist circumference [104]. Despite a growing body of literature on the topic, the relationship between neighborhood greenery and body weight is inconsistent due to the variety of systems used to evaluate green space and health status and the inconsistent results obtained from different evaluation criteria in different research areas [34].

Some studies have found that living closer to green environments is associated with recommended body mass index (BMI) values [55,56,95]. In regard to blue spaces, an 8-year study by Halonen, et al. [105] found that living further from waterfronts in urban areas increases the risk of overweight. However, other investigations found no such association [49,106,107]. Most of these studies assessed adiposity levels through self-reported BMI, weight, and height. To the best of our knowledge, there are few studies on this topic that only focus on women. It is important to note that some studies have shown that self-reported BMI and measured BMI have a good correlation [108], with individuals generally overestimating their height and underestimating their weight. Thus, it is not appropriate to use cut-off scores for obesity based on self-reported data, but rather raw BMI scores. This may partially explain the inconsistent results in previous studies.

According to data from the World Obesity Federation Global Obesity Observatory (accessed on 20 December 2022)), approximately 32% of adult women in Portugal are considered obese. In our study, most of the participants had normal levels of total and central adiposity, which might be due to the fact that a large percentage of the women were under the age of 45. Menopause typically occurs between the ages of 45 and 55, and, regardless of age, estrogen depletion is associated with an increase in fat mass, especially in the abdominal region [18].

In our research, total and central adiposity levels were assessed considering octopolar bioimpedance, and we found that women who lived closer to green spaces exhibited better levels of total and intra-abdominal adiposity. As in the present investigation, no association was identified between distance to green spaces and the levels of physical activity, we cannot assume these as mediators of the relationship. We therefore hypothesize that our results may be related to specific environmental factors that may have contributed to normal weight values, such as the diversity of fauna and flora in the study area. In line with some previous research, the presence of native plants [109], a variety of plant species [110,111], and grasslands [112] contribute to the control of adiposity levels.

Furthermore, the existence of wildlife—fish, birds, horses, butterflies, and others—also contributes to normal values of weight [109]. Additionally, within this domain, natural spaces that provide room and infrastructure for the practice of equestrian [113] and/or aquatic activities [109], cycling [114], or running [115] and the existence of outdoor recreation resources [116,117] are also some of the factors that contribute to better weight control. These factors, although not directly measured in our research, could potentially explain our results. Some studies, e.g., [118], have suggested that contact with natural environments, such as forests, increases adiponectin levels and reduces the risk of inflammation and atherosclerosis. It would have been valuable to collect information on the duration and frequency of exposure to nature in our research.

In relation to blue spaces, no association was recorded between living close to them and body fat levels. The limited availability of blue spaces in the study area, and the lack of opportunities for physical activity, could explain the lack of association recorded between living close to them and body fat levels. Elliot et al. [102] suggest that individuals living near the coast are healthier than those living inland, showing that the coastal environment may not only offer better opportunities for its inhabitants to be active, but also provide significant benefits in terms of stress reduction.

Another health outcome that has been associated with the residential environment is sleep. An Australian study found that people who live near green spaces exhibited better sleep quality [27]. Similarly, Grigsby-Toussaint, et al. [119] reported that access to green spaces has health benefits through increased exposure to natural daylight patterns, helping to maintain circadian rhythms. Triguero-Mas, et al. [120] also documented that the methods of global position system (GPS) trajectory measurement, which capture the actual daily green space exposure, had a positive association with sleep quality, while the residential-area-based measurement method does not always have a statistically significant association. In our research, we had no data related with the use of green space, which could have been interesting. In a review conducted by Shin, et al. [121], associations of green space with improved sleep outcomes were shown for studies in which participants’ use of green space (gardening or walking) was assessed as well as in studies that only evaluated study participant surroundings (accessibility and availably). In our study, women who lived closer to green environments (and not blue spaces) had shorter periods of sleep onset latency, which is a parameter of sleep quality. Surprisingly, the distance from natural environments was not related to sleep duration.

In our research, sleep was objectively measured using accelerometers. Although there have been validation studies of accelerometers in the assessment of sleep [122], questionnaires are the most commonly used method for measuring sleep quality and quantity [121]. In a systematic review of thirteen studies conducted by Shin, Parab, An and Grigsby-Toussaint [121], only two evaluated sleep patterns using accelerometry. We therefore consider that the variables effectively measured and self-reported may contribute to the differences found in the results.

Another explanation for our findings regarding sleep onset latency is related to the effect of green spaces on mood and mental health and the reduction in noise pollution [123,124]. The latter may contribute to a delay in sleep onset latency and has been associated with nocturnal awakenings [125].

The literature suggests a strong link between stress levels and adiposity levels [126,127], and it is known that high levels of intra-abdominal adiposity can lead to poor sleep quality. Our study found that most of the women had normal levels of total and central adiposity, which may help to explain the positive results we observed regarding sleep onset latency. However, we did not measure stress levels in our research.

We anticipate different outcomes would result if we were to differentiate between women based on their age and reproductive stage. According to some authors [128,129], sleep patterns are known to vary with age due to changes in circadian and homeostatic processes, as well as normal physiological and psychosocial changes. The reproductive aging stage, especially the transition to menopause, also has an impact on women’s sleep. The presence of vasomotor symptoms (such as hot flashes and night sweats) resulting from estrogenic depletion, commonly in conjunction with low mood and anxiety, leads to sleep difficulties [130,131,132]. Our hypothesis is that if we account for these variables, we may be able to identify correlations between sleep duration and other factors.

Our findings can also be attributed to the fact that the women in our study, with an average age of 37.78 ± 14.70 years, were still able to produce recommended levels of melatonin for the regulation of circadian rhythms. According to Karasek [133], melatonin secretion generally decreases with age and becomes significantly lower in most people by the sixth or seventh decade of life [134]. This could be another reason why exposure to green spaces did not have a positive effect on sleep duration in our study.

The lack of a correlation between blue spaces and sleep duration in our study may be due to the fact that the study area consists of rivers with low flow. According to Liu, et al. [135], the positive impact of blue spaces on sleep duration and quality is more prominent in coastal areas, which are considered restorative due to the high concentration of negative ions. In a meta-analysis conducted by Georgiou, et al. [136], the authors found that proximity to blue spaces or a higher concentration of blue spaces generally had a positive effect on restoration in adult populations, compared to living further away or in areas with a lower concentration of blue spaces. The restorative process triggered by blue spaces could be a key factor in promoting better sleep patterns.

The absence of a connection between distance to green and blue spaces, physical activity levels, and sleep duration in our study could also be due to the fact that people who live in these environments are already physically active and have good sleep habits. According to Goldenberg et al. [137], people with higher incomes tend to have greater access to nearby green and blue spaces that are within walking distance of their homes. Furthermore, wealthy residential areas tend to have abundant high-quality green spaces, while lower-income neighborhoods often lack such spaces [138]. According to Markevych et al. [23], the health benefits from green spaces can be even more crucial for people with lower socioeconomic status (SES) and those who live in more deprived neighborhoods. People with a lower SES typically have poorer baseline health, limited mobility, and live in areas with higher pollution, which can be addressed through experimental studies, although these are challenging to implement in practice.

The strengths of this study include the use of geocoding to accurately determine individuals’ proximity to natural environments, the use of objectively measured body composition to avoid potential biases associated with self-reported anthropometric measures, and the use of triaxial accelerometers to objectively measure physical activity levels. Additionally, the study sample was limited to women, which is another strength.

Despite the strengths explained previously, several limitations of this study need to be acknowledged. First, our sample size was relatively small, which limits our ability to make inferences and extrapolate our results. It would have been beneficial to have a larger sample size, which would have allowed us to analyze results between groups based on factors such as age or reproductive stage. Second, this study could have also benefited from a more equally adjusted number of demographic variables, such as marital status. Third, the study did not evaluate the use of natural spaces for activities such as gardening, physical activity, or relaxation. Additionally, the proximity to natural environments was assessed without considering differences in the types, sizes, or qualities of these spaces, which could result in a substantial degree of heterogeneity in exposure to recreational opportunities. To better understand the characteristics of these environments that can encourage physical activity and improve sleep and body composition, future research could assess cognitive variables such as knowledge about different species. According to Vanhöfen et al. [139], it is important to understand how people perceive their natural surroundings.

## 5. Conclusions

To the best of our knowledge, this study is unique in that it is the first to examine the relationship between physical activity, sleep, body composition, and proximity to natural spaces specifically in women. Our findings suggest that proximity to green spaces is associated with more favorable levels of total and central adiposity and lower sleep onset latency in adult women. However, we did not observe any relationships with health indicators and proximity to blue spaces. Further research is needed to better understand the nature of this relationship, particularly by exploring different metrics of nature exposure. Additionally, future studies should consider analyzing the specific characteristics of these natural environments to inform their design and optimize their use.

## Figures and Tables

**Figure 1 ijerph-20-03647-f001:**
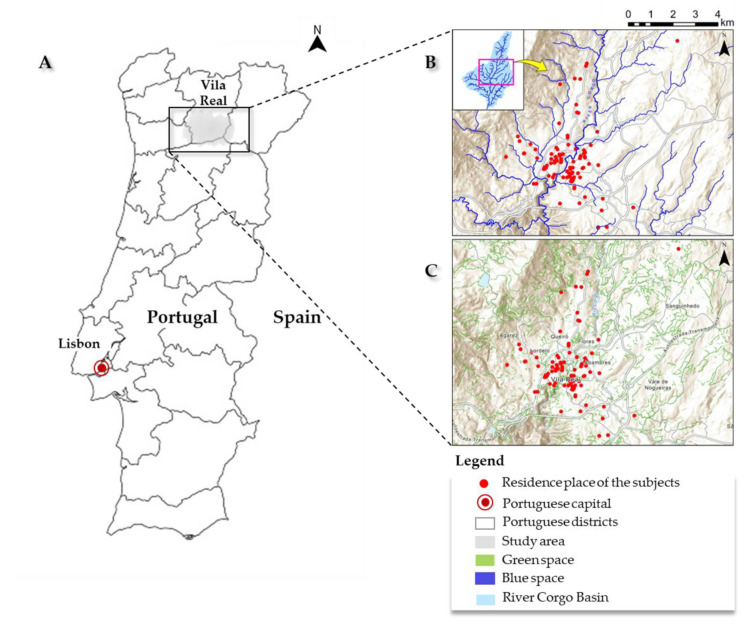
Location of the study area in Portugal (**A**), with blue (**B**) and green (**C**) areas.

**Figure 2 ijerph-20-03647-f002:**
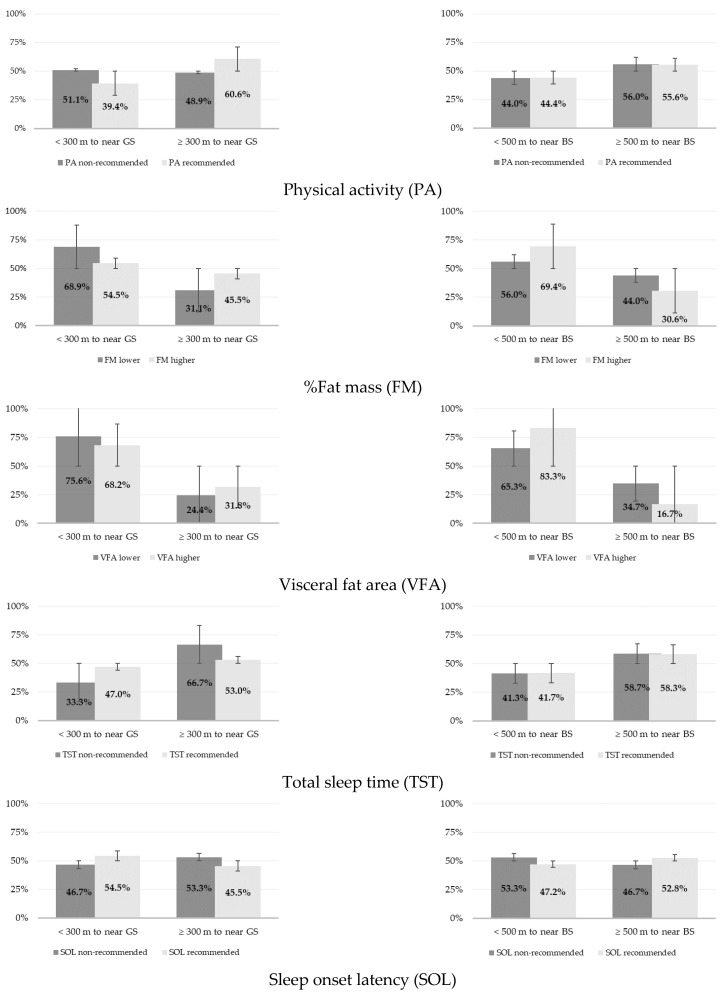
Classification of physical activity levels, adiposity, and sleep outcomes according to distances to green (300 m) and blue spaces (500 m); GS—green space; BS—blue space.

**Figure 3 ijerph-20-03647-f003:**
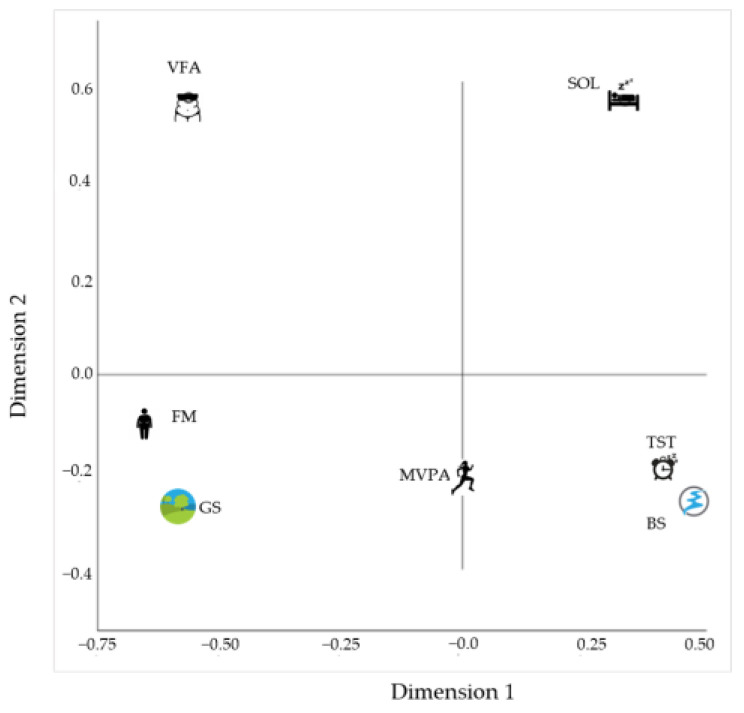
Component loadings.

**Figure 4 ijerph-20-03647-f004:**
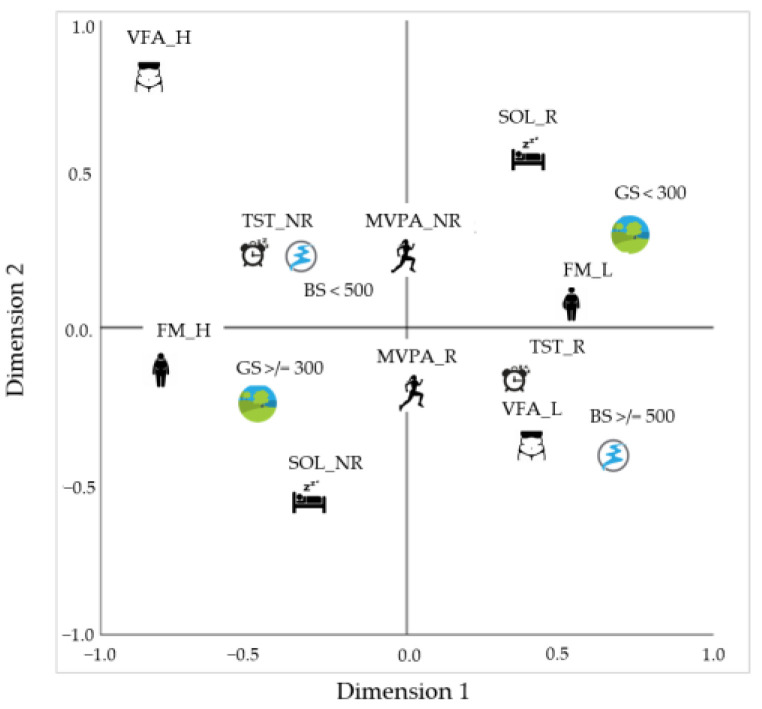
Centroid plot.

**Figure 5 ijerph-20-03647-f005:**
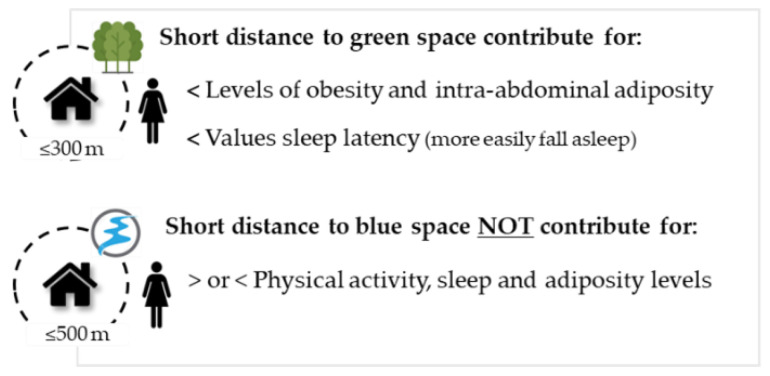
Main results of study.

**Table 1 ijerph-20-03647-t001:** Baseline characteristics of study participants.

Variables	(*n* = 111 Women)
	Mean ± SD
Age (years)	37.78 ± 14.70
Body composition	
Body mass (kg)	61.02 ± 8.83
Height (m)	1.63 ± 0.06
Fat mass (kg)	18.59 ± 6.81
Fat mass (%)	29.61 ± 73.39
Visceral fat area (cm^2^)	79.09 ± 34.97
Appendicular skeletal muscle mass (ASMM, kg)	17.26 ± 2.58
Appendicular skeletal muscle mass index (ASMMI, kg/m^2^)	6.51 ± 0.69
Physical activity (PA)	
Total PA (min/week)	745.98 ± 313.50
Moderate-to-vigorous PA (min/week)	188.03 ± 118.99
Steps/day (*n*)	12,712.98 ± 3816.90
Sleep	
Total sleep time (h)	7.04 ± 1.18
Sleep efficiency (%)	93.20 ± 2.83
Sleep onset latency (min)	32.11 ± 13.68
Nocturnal awakenings (*n*)	15.15 ± 5.31
Minutes of awakenings (min)	2.09 ± 0.44
Sleep fragmentation (%)	23.14 ± 8.78
Sleep fragmentation index (*n*)	3.51 ± 1.67
Sleep onset time (HH:MM ± min)	00:27 ± 30
Sleep offset time (HH:MM ± min)	08:05 ± 10
Sleep midpoint time (HH:MM ± min)	04:19 ± 02
Distance to natural spaces (m)	
Green spaces	360.76 ± 244.30
Blue spaces	421.45 ± 207.83

SD—standard deviation.

**Table 2 ijerph-20-03647-t002:** Classification of adiposity, physical activity levels, and sleep (*n* = 111).

Variables	*n* (%)
Fat mass (%)	
Non-obese	67 (60.4)
Obese	44 (39.6)
Visceral fat area (cm^2^)	
Normal	79 (71.2)
High	32 (28.8)
Appendicular skeletal muscle mass index (kg/m^2^)	
Normal	101 (91.0)
Deficit	10 (9.0)
Moderate-to-Vigorous PA (min/week)	
Non-recommended	49 (44.1)
Recommended	62 (55.9)
Total Sleep Time (min)	
Non-recommended	46 (41.4)
Recommended	65 (58.6)
Sleep efficiency (%)	
Non-recommended	---
Recommended	111 (100.0)
Sleep onset latency (min)	
Non-recommended	57 (51.4)
Recommended	54 (48.6)
Sleep fragmentation index (event/h)	
Normal	92 (82.9)
Possible sleep disturbance	19 (17.1)
Sleep midpoint	
Before 2:00 AM (early)	2 (1.8)
Between 2:00 and 4:00 AM (healthy)	42 (37.8)
After 4:00 AM (late)	67 (60.4)

**Table 3 ijerph-20-03647-t003:** Two-dimensional results.

	Sets	Dimensions		Sum
		Dimension 1	Dimension 2	
Loss	Distance to natural spaces	0.484	0.826	1.310
	Actigraphy variables	0.708	0.559	1.267
	Body composition variables	0.526	0.407	0.933
	Mean of sets	0.573	0.597	1.170
Eigenvalue		0.427	0.403	
Fit				0.830

## Data Availability

Not applicable.

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
