# Peer review of "Distance to Natural Environments, Physical Activity, Sleep, and Body Composition in Women: An Exploratory Analysis"

_ijerph, 2023, doi:10.3390/ijerph20043647_

Round 1
Reviewer 1 Report
This study evaluated accessibility to green and blue spaces using a geographic information system-based method. Physical activity and sleep parameters were measured using ActiGraph accelerometers and body composition was assessed using octopolar bioimpedance (. Nonlinear canonical correlation analysis was used to analyze the data. Authors revealed that women living in green spaces close to their homes had lower levels of obesity, intra-abdominal adiposity and have shorter sleep latency. No relationship was found between physical activity and sleep duration. In relation to blue spaces, the distance to these environments was not related to any healthy indicator analyzed in this study.
To me, it is clear that the authors are rather new to scientific research, and need more time to read up on the subjects to be able to make the mandatory changes to the manuscript for it to be publishable. The authors need to making a correct statistical analysis of their data.
I would recommend that the authors withdraw the manuscript from the publication procedure and follow some of the advice above, before attempting to publish.
• One general issue has to do with the quality of the writing. I would encourage them to enlist the help of a native English speaker with some familiarity with the research area so they can tighten the grammar.
• Also, from the introduction, I did not found any necessity to do this study? No study focused on women is not reasonable, the author should explain why lack of study on women is a problem.
• In introduction, the authors did not mention why they need the sleep data, and the relationship between sleep and health was not mentioned.
• line 43. “women remain one of the marginalized and vulnerable groups in society, often exhibiting worse health outcomes compared to men.” WHO data shows that average life expectancy of women in Portugal is longer than men, why did you think their health condition is worse than men? Please explain, and show some references.
• line 53-54. “In relation to sleep, women typically report poorer quality and more disrupted sleep across various stages of life.”. Reference needed.
• Line 56. “Regarding body composition, in the literature, the differences between genders are well established.” Consider present study mainly used physical parameters, not “gender” but “sex” should be used.
• line 66. “as promoters of health and well-being.” Reference needed.
• line 69. “heat”. Please explain why heat was considered as urban environmental hazards? Did you mean extreme heat, or heat wave, or heat island?
• line 80-82. “T According to Völker and Kistemann [21], blue spaces are independent entities and there is a need to consider them separately and not solely as a subcategory of green spaces.” The authors should explain or at least quotewhat Völker and Kistemann said in their study.
• line 83, 85. “Residential environment” and “neighborhood environment” are different things. Which one did you study?
• line 85. “greater exposure to their neighborhood environment and are more vulnerable to its effects” what effect?
• line 87. “mandatory”? did you mean “necessary”?
• line 87. “until now we have not identified any research that examined this issue exclusively in women.” What kind of issue? The author did not mention.
• line 95. What is “Csb”? please use full form when it first appeared.
• line 98-110. what is the different between green and blue space? it seems blue space also have abundant terrestrial plant.
• line 100-106. Italic style should be used when you write scientific name of plants.
• line 100-106. What is the “localized plants”?
• Figure 1. 1)what does the subject (red dot) mean? 2) Corgo River basin could not be find on the map.
• line 214-217. Authors need to explain why they set “300 meters of green and 500 meters of blue space respectively.” ?
• line 239-240. “Of the women who lived closer to green spaces, 47% sleep at least 7h per night and, in comparison, those who live further away, 66.7% exhibited non-recommended hours of sleep.” I wonder if distance to green space can influence people’s sleep length rather than their distance to work place or their cell phone habits. Please explain.
• Figure 2. 1) absent of error bar; 2) absent of Y axes.
• line 249-256. There was not any data showed about your principal component analysis. please report the crucial data in text or make a table. Figure 3 is not enough at all.
• line 250,251. The abbreviations, such as (SL_R) and (FM_L) only appeared once in the whole manuscript, are they necessary?
• line 263. “European Union countries (47%), including Portugal (52%)” what are these data mean? Where are they come from?
• line 378. “Elliott, White, Grellier, Rees, Waters and Fleming [69] suggest” citation style is strange in here and many other places.
Author Response
Dear Reviewer,
I am writing to express my sincerest gratitude for taking the time to review my manuscript. Your comments and suggestions were invaluable in improving the quality of the work. I appreciate your insights and the effort you put into providing a thorough evaluation.
I have made several revisions to the manuscript based on your suggestions, and I believe that the manuscript is now stronger as a result.
I would like to thank you again for your time and expertise in reviewing this work. Your contributions are greatly appreciated.
Sincerely,
Andreia Teixera

Reviewer 2 Report
I like the physiological approach. Probably you can introduce new variables, such as chronotype, based on the midpoint of sleep which could be calculated from your actigraphy data? Then you may be able to consider also the chronotype/morningness-dimension and make the paper stronger.
Line 52: it is not all about sleep quality but you should also cite a paper about sleep length which is dependent on age, longer or shorter when compared to men.
Line 62: probably there are differences in BMI? As far as I know there are more men obese or with an higher BMI than women in many countries which counteracts your argument.
Line 75: just a notice: this is a very convincing information of your study.
The part about green spaces / blue spaces is a bit thin and might be improved by an additional paragraph to analyse in more details which factors are important, especially the typical studies of Dallimer et al., (2012) and of others in that direction. Dallimer, M., Irvine, K. N., Skinner, A. M., Davies, Z. G., Rouquette, J. R., Maltby, L. L., ... & Gaston, K. J. (2012). Biodiversity and the feel-good factor: understanding associations between self-reported human well-being and species richness. BioScience, 62(1), 47-55.
Line 190: some citations/references are missing that support your statements (they are correct, but needs some citations)
Line 210: is the high amount of singles typical for this area Vila Real? If not, this should be discussed in the limitations section - I feel that the value is too high given the mean age of your sample – in other areas this is the typical reproductive phase with one to three children…
Also a citation is necessary on what you base the recommended hours of sleep.
Line 245: sleep latency or sleep onset latency? Please check which is the correct term.
Line 328: of course, because nearly all blue spaces are surrounded by green, in my view it cannot be separated because all spaces are green, but some have additional blue.
One major concern with all of these studies is that it is not the environment that impacts health, but that healthier people (or/and wealthier people) choose a living near green spaces because these people are already really active and want to pursue their activity near from home, suggesting that they choose their living there. This must be discussed as a major concern that cannot be solved (except with experiments, which are nearly impossible).
Line 345: there are some studies that show that self-report BMI and BMi measured afterwards show a good correlation, with “all” people overestimating their height and underestimating their weight. Thus, cutoff scores for obesity should not be used in self-report, but raw BMI scores. This might be a reason why other studies were not good enough. However, you have measured these variables objectively. Perhaps this can be added here.
One last thing that is not discussed here are cognitive variables. When it comes to wildlife, you mention some studies that report an effect of wildlife / species richness on well-being – in some cases, this can be related to knowledge. For example Vanhöfen, J., Schöffski, N., Härtel, T., & Randler, C. (2022). Are Lay People Able to Estimate Breeding Bird Diversity?. Animals, 12(22), 3095. found that lay people are able to estimate bird species richness, which in turn, can add to the non-cognitive aspect.
Author Response
I want to thank the reviewer for taking the time and effort to review the manuscript. We sincerely appreciate all your valuable comments and suggestions, which helped us improve the manuscript's quality.
